# Multi-View Stereo Vision Patchmatch Algorithm Based on Data Augmentation

**DOI:** 10.3390/s23052729

**Published:** 2023-03-02

**Authors:** Feiyang Pan, Pengtao Wang, Lin Wang, Lihong Li

**Affiliations:** School of Information and Electrical Engineering, Hebei University of Engineering, Handan 056038, China

**Keywords:** multi-view stereo, data augmentation, patchmatch, adaptive propagation

## Abstract

In this paper, a multi-view stereo vision patchmatch algorithm based on data augmentation is proposed. Compared to other works, this algorithm can reduce runtime and save computational memory through efficient cascading of modules; therefore, it can process higher-resolution images. Compared with algorithms utilizing 3D cost volume regularization, this algorithm can be applied on resource-constrained platforms. This paper applies the data augmentation module to an end-to-end multi-scale patchmatch algorithm and adopts adaptive evaluation propagation, avoiding the substantial memory resource consumption characterizing traditional region matching algorithms. Extensive experiments on the DTU and Tanks and Temples datasets show that our algorithm is very competitive in completeness, speed and memory.

## 1. Introduction

The goal of multi-view stereo (MVS) is to recover the 3D information of objects from a collection of images with known camera parameters, which is one of the most challenging tasks in computer vision. Although after decades of development, MVS has made a major breakthrough in geometry and can realize the mapping from two-dimensional pictures to three-dimensional models, it is still, however, a challenge for MVS to recover the three-dimensional information of objects from images for actual situations such as image occlusion, changes in lighting conditions, non-textured areas, or non-Lambertian planes [1,2]. In 2015, the success of convolutional neural networks (CNN) in image processing, brought a breakthrough for this challenge of MVS.

Learning-based MVS can extract more feature information from images by CNN to perform convolution operations on images, which can improve some problems that cannot be handled in traditional MVS. Indeed, many learning-based algorithms [3,4,5,6,7] have superior performance on weakly textured regions, reflections, etc. These algorithms fuse the feature information of the reference image and the source images to construct 3D cost volume and then calculate the expectation on each depth value of the 3D cost volume to predict the final depth map. They realize the construction of 3D cost volume through 3D convolution, and 3D convolution needs to consume a lot of computing resources and memory resources. In order to save the amount of calculation, some algorithms [4,8,9] are proposed to utilize classification methods to construct the cost volume. The probability of each depth value is predicted from the 3D cost volume, and the depth layer with the maximum probability is utilized as the final predicted depth value. Although such algorithms cannot directly infer the exact depth value from the model, they can improve the robustness of the network by constraining the cost volume through the cross-entropy loss operation on the confidence volume [4].

Although the above achievements have achieved great success in the MVS test indicators, the wrong feature information will still be extracted from the highlighted areas of the image. Therefore, it will generate wrong guidance for the final depth map construction, and ultimately affect the final reconstruction results [3]. We can adopt a discarding strategy for the highlighted area of the image to avoid wrong guidance. In [10], the strategy of data augmentation was applied to the self-supervised MVS network to reduce the error of the photometric consistency assumption in the actual process and achieved good results. Different from the traditional planar sweeping algorithm, the patchmatch algorithm utilizes a random iterative algorithm to calculate the nearest neighbor and obtains the optimal depth through spatial correlation without predicting all depth ranges. In particular, Wang et al. proposed PatchmatchNet [7] based on this strategy. Since there is no need to calculate each depth probability value, the calculation consumption is greatly reduced and the performance is good.

In this paper, we propose a data-augmentation-based patchmatch algorithm for multi-view stereo (Data-Augmentation PatchmatchNet). In the data input stage, the robustness of the network to the image is enhanced by changing the brightness, contrast, and hue of the input image and randomly removing some regional pixels from the image. In the depth estimation stage, the method of *dynamic interval d* is adopted to obtain more image information. In addition, we also utilize the feature information of the neighborhood pixel to construct the cost volume of the pixel through the patchmatch algorithm, thus completing the optimal estimation of the depth value. The wrong feature information was extracted due to image problems. The depth map reconstructed from the reference frame image not only contains the feature information of the reference frame image, but also can extract the feature information of the scene from the source images. The features of multiple images are fused to obtain the depth map of the reference frame, so as to avoid misleading the final reconstruction result due to the wrong information about a single image. More importantly, the data augmentation strategy can greatly simulate the error information contained in a single picture, making the network more robust.

## 2. Related Work

### 2.1. Traditional MVS

Traditional MVS algorithms can be broadly classified into four categories: voxel-based [11,12], surface estimation-based [13], region estimation-based [14], and depth map-based [15,16,17] algorithms. Depth maps are more favored due to their better flexibility and scalability. Gipuma et al. [15] proposed a propagation scheme based on the black-and-white checkerboard. This scheme is conducive to parallel computing, and the PatchMatch stereo method is applied to multi-view reconstruction. COLMAP was proposed by Schönberger et al. [16], which utilized joint estimation of pixel view, the depth map, and the normal vector to improve the performance of the model. ACMM proposed a multi-scale MVS framework with adaptive checkerboard propagation and multi-hypothesis joint view selection [17]. This idea based on patchmatch inspired us to propose patchmatch based on deep learning.

### 2.2. Learning-Based MVS

Compared with traditional methods, learning-based multi-view stereo vision can extract more feature information from images, thus achieving better reconstruction results. In recent years, with the development of convolutional neural networks, many excellent learning-based multi-view stereo vision algorithms have been proposed. SurfaceNet [18] adopts a 3D convolutional network constructed by encoding camera parameters and images in voxels. In contrast, Yao et al. [3] proposed a more broadly adaptable approach to deep learning. Specifically, MVSNet performed feature extraction for each image input into the network. The cost volume was obtained by fusing the image features of the reference image and the source images through the camera parameters. Finally, 3D convolution was utilized to regularize the cost volume containing depth assumption information, and the final predicted depth map was obtained by regression. Due to 3D convolution, MVSNet has high memory and computing requirements; therefore, it can only reconstruct images with smaller resolutions, which does not meet the reconstruction needs of large scenes. To solve this problem, R-MVSNet [4], proposed by Yao et al., utilized the GRU structure in the recurrent neural network to regularize the cost volume, which can reduce the need for computing and memory, but increase the running time of network training. IterMVS [19] encodes the pixel level of depth through the GRU estimator to obtain a higher-precision depth map. Study [20] adopts a hybrid Long Short-Term Memory (LSTM) for cost volume regularization to replace the traditional 3D CNN. D2HC RMVSNet [21] is an efficient dense hybrid recursive multi-view stereo network with dynamic consistency check. Fast-MVSNet [6] adopts a coarse-to-fine network structure, which can quickly and accurately perform multi-view depth estimation. Inspired by this, a coarse-to-fine strategy is often utilized to improve the speed and accuracy of the network. CVP-MVSNet [22] utilizes image pyramids to obtain feature maps of different scales, and obtains residual depth maps by reducing the remaining depth range. CasMVSNet [23] is similar to CVP-MVSNet in the processing process; however, the range of the residual depth map is obtained by the re-projection of pixels. UCS-Net [24] adopts the strategy of assuming depth uncertainty at each stage to optimize the depth map. WT-MVSNet [25] adopts a cost transformer (CT) instead of 3D convolutions for cost-volume regularization to reduce memory consumption.

In addition, AttMVS [26] utilizes a novel attention-enhanced matching confidence module and attention-induced regularization module to improve the robustness of the network. Vis MVSNet [27] infers and integrates occluded information in MVS networks by matching uncertainty estimation strategies. AA RMVSnet [28] is an adaptive aggregation module and utilizes a hybrid network with a recurrent structure for cost volume regularization. PatchmatchNet [7] introduces traditional region matching algorithms into the deep learning framework, which greatly reduces the running time and memory consumption. Compared with PatchmatchNet, our algorithm proposed in this paper can reconstruct a more complete model.

## 3. Proposed Algorithm

In this section, we will introduce the network structure of our DA-PatchmatchNet in detail, as shown in Figure 1. It mainly includes a depth estimation branch and data augmentation branch. In the depth estimation branch, we utilize image pyramids to extract image information, take advantage of patchmatch and adaptive cost aggregation to construct cost volume, and finally regress the predicted depth map. In the data augmentation branch, we adjust the brightness, hue, saturation and other information of the image, and randomly discard some pixels to reduce the influence of image reflection areas on the reconstructed depth map. Subsequently, this section will describe multi-scale feature extraction, learning-based patchmatch, sampling interval d, data augmentation and loss calculation.

### 3.1. Multi-Scale Feature Extraction

Input images to the network whose length and width are W×H respectively. These images includes a reference image $I_{0}$ and N−1 source images ${\left\{I_{i}\right\}}_{i=1}^{N-1}$. Before inputting to the patchmatch network, we adopt a strategy similar to a feature pyramid network (FPN) [13] to extract features of different scales from the image. Feature extraction is performed on images of different scales in a coarse-to-fine fashion to optimize network performance. Specifically, we employ three scaling-stage image pyramids to extract features (k=1,2,3). $F_{i}^{k}$ represents the feature map of image $I_{i}$ at the k stage, and its output size is $\frac{W}{2^{k}}\times \frac{H}{2^{k}}$.

### 3.2. Learning-Based Patchmatch

Combining traditional region matching and learning-based depth estimation algorithms, we improve on PatchmatchNet [7]. Next, we will introduce the initialization of the depth map, adaptive propagation, depth interval, adaptive evaluation, and depth map regression.

#### 3.2.1. Initialization and Local Perturbation

In the first region matching, since the network does not have an initial depth map, the depth map needs to be initialized. Initialization is performed in a random way.

Within the depth range $\left[d_{min},d_{max}\right]$ of each frame image, we uniformly sample the assumed image depth $D_{f}$ of each image pixel to ensure the diversity of initialization. Such an approach facilitates the application of our model to large-scale scene reconstruction [4,29].

In addition, after the initial depth map is available, we introduce local perturbation to continuously optimize the depth map in the iteration of the subsequent stage k. Specifically, we increase the local perturbation by making a depth hypothesis for each pixel $P_{k}$ in the depth range $R_{k}$ of the original depth map and at smaller stage k. We decrease the range $R_{k}$ for finer refinement of performance. For the depth map of each stage, we iterate multiple times to achieve the optimal effect. When it is the input of the next stage, we will upsample it so that the depth map can be optimized on a larger scale.

#### 3.2.2. Adaptive Propagation

The depth value of the object in the image has a certain spatial correlation; however, it is only expressed by the pixels on the surface of the object. We adopt the strategy of PatchmatchNet to achieve adaptive propagation of neighborhood pixels. This differs from DeepPruner [30], which predicts pixel depth only through simple neighborhood pixel depth propagation. As shown in Figure 2, the spatial correlation of the pixel depth value of the object surface can be obtained to the maximum extent through this adaptive propagation, which makes the model converge to a more accurate depth value quickly.

Specifically, adaptive propagation of the network is realized through a variable convolution. $K_{p}$ represents the predicted depth value corresponding to the reference image pixel p. The 2D additional offsets are ${\left\{\Delta o_{i}\left(p\right)\right\}}_{i=1}^{K_{p}}$, which are learned by the model. The additional offsets are applied to the fixed offsets ${\left\{o_{i}\right\}}_{i=1}^{K_{p}}$ to form our internet.

For the predicted depth $D_{p}\left(p\right)$, the feature map of the reference image obtained by multi-scale feature extraction is input into the 2D convolutional network. Then the pixel offset is learned, and finally obtained through bilinear interpolation. The process can be described by Equation (1).

For the predicted depth $D_{p}\left(p\right)$, we feed the feature maps obtained by multi-scale feature extraction from the reference image into a 2D convolutional network. Then the algorithm learns the pixel offset, which is finally obtained by bilinear interpolation. The process can be described by Equation (1).
(1)$$D_{p}\left(p\right)={\left\{D_{p}\left(p+o_{i}+\Delta o_{i}\left(p\right)\right)\right\}}_{i=1}^{K_{p}}$$
where D is obtained by upsampling the depth map output by the previous iteration.

#### 3.2.3. Adaptive Evaluation

Feature mapping

The current learning-based MVS algorithms [3,4,6,23] all adopt a strategy similar to the planar scanning algorithm in the traditional algorithm. The image depth is $\left\{d_{j}|j=1,\ldots ,D_{f}\right\}$. A fixed number of depth layers is sampled from front to back within the depth range $\left[d_{min},d_{max}\right]$. The image features of the reference image are mapped to the depth layer. In addition, the features of the source images are also mapped to the depth layer using the camera parameters of the reference image and source images. For each pixel p of the reference image, the feature $F_{i}$ of the source images is mapped to the assumed depth $d_{j}$ of the jth layer. The formula is as follows:(2)$${p^{\prime }}_{i,j}=K_{i}\cdot \left(R_{0,i}\cdot \left(K_{0}^{-1}\cdot p\cdot d_{j}\right)+t_{0,i}\right)$$

$K_{i}$ represents the intrinsic matrix of the reference image. $R_{0,i}$ and $t_{0,i}$ represent the rotation matrix and translation matrix of the source images and the reference image, respectively. After MVSNet [3], the ${p^{\prime }}_{i,j}$ and the features $F_{i}$ of the reference image are obtained. The fused $F^{\prime }$ is obtained by bilinear interpolation. $F^{\prime }$ is the feature set of the corresponding pixels of the reference image and the source images.

Cost matching

In multi-view stereo vision, cost matching is to fuse the information of a specified number of neighboring frame images Iii=1N−1 and reference image I0 into a single pixel p and hypothetical depth dj. The pixel view weights are obtained from the smallest scale image. Feature fusion is performed through the pixel view weights. F0p and Fipi,j∈ℝH×W×C represent the features obtained from the reference image and the source images through the feature extraction network, respectively. Their similar features are expressed as:(3)Sip,j=F0p,Fipi,j

⋅,⋅ represents the inner product. Sip,j represents the similarity of the corresponding pixel features of the reference image and the source images. The tensor Sip,j is Si∈ℝW×H×D×C.

The strategy of pixel view weight is adopted here. Pixel view weight represents the pixel visibility information of the reference image and the source images. wipi=1N−1 represent the pixel view weights. wip is only obtained by initializing the depth assumption in the first iteration of the smallest scale, which is calculated only once. The more refined information of the image is obtained by continuous upsampling. 3D convolution builds a network for extracting pixel view weights, which consists of 1×1×1 convolution kernels without bias and sigmoid layers. The similarity set Si corresponds to a number between 0 and 1, which represents a pixel output. Pi∈ℝW×H×D represents the confidence of the pixel corresponding to the depth. The specific formula is as follows:(4)wip=maxPip,jj=0,1,…,D−1

$P_{i}\left(p,j\right)$ represents the visibility confidence of pixel p at the j-th layer of the hypothesized depth.

S¯p,j represents the set of feature similarities between the final reference image and source images, which is obtained by weighted sum of Sip,j and wip.
(5)S¯p,j=∑i=1N−1wip⋅Sip,j∑i=1N−1wip
where S¯p,j∈ℝW×H×D×C. The feature similarity set is converted into a matching cost C∈ℝW×H×D by a 3D convolution which contains a 1×1×1 convolution kernel.

Adaptive Spatial Cost Aggregation

Similar to adaptive propagation, regular windows are utilized to realize cost aggregation in the traditional MVS matching algorithm. However, this regular window is not conducive to the stability and smoothness of cost aggregation. Therefore, PatchmatchNet’s adaptive cost aggregation strategy is applied to improve the deficiency. ${\left\{p_{k}\right\}}_{k=1}^{K_{e}}$ is proposed for each pixel p, where $K_{e}$ is a spatial window. When learning, the offset of each pixel under the spatial window is ${\left\{△p_{k}\right\}}_{k=1}^{K_{e}}$. The cost aggregation is defined as:(6)$$\tilde{C}\left(p,j\right)=\frac{1}{\sum_{k=1}^{K_{e}}w_{k}d_{k}}\sum_{k=1}^{K_{e}}w_{k}d_{k}C\left(p+p_{k}+△p_{k},j\right)$$

$w_{k}$ represents the spatial similarity. In the reference image feature map, the feature similarity of the surrounding position of pixel p is sampled, then the normalized weight of the similarity between the pixel and the sampling point through 3D convolution is output. Then we obtain the normalized weight of the similarity between the pixel and the sampling point through 3D convolution. $d_{k}$ represents the depth hypothesis similarity. The absolute value of the sampling point and pixel p hypothetical depth is obtained, then it is normalized and a sigmoid operation is performed to output the depth weight finally.

Depth Regression

Similar to MVSNet, softmax is applied to convert the cost volume obtained by adaptive cost aggregation into a confidence volume. Finally, the depth map is obtained by seeking the expectation.
(7)$$D_{E}\left(p\right)=\sum_{j=0}^{D-1}d_{j}\cdot P\left(p,j\right)$$

### 3.3. Dynamic Interval d

In most learning-based MVS, the depth sampling range is set to $\left[d_{min},d_{max}\right]$. The number of sampling depth layers is fixed within the depth range, then the sampling interval is determined. However, in multi-scale feature extraction, a fixed sampling interval for different scale images will lose the feature information of the image. Therefore, we adopt a strategy of dynamic sampling intervals. The sampling interval is determined according to the image pixel difference at different stages. The specific processing method is shown in Figure 3. By setting different pixel differences, the sampling interval is calculated by the parameters of the image and camera. The calculation formula is as follows:(8)$$\Delta d=K_{i}\cdot \left(R_{0,i}\cdot \left(K_{0}^{-1}\cdot \Delta {p^{\prime }}_{i,j}\cdot d_{j}\right)+t_{0,i}\right)$$

$\Delta {p^{\prime }}_{i,j}$ is the difference between adjacent pixels on the projected view. The interval *d* is only updated when the image scale changes.

### 3.4. Data Augmentation

In recent years, some research results have shown that data augmentation [10,31,32] has a positive effect on learning-based reconstruction results. The intuitive understanding is that data augmentation brings more abundant samples to training, thus influencing the output of the results. In fact, data augmentation brings some characteristics to the model that the original training data does not have. This improves the robustness and generalization ability of the model. Specifically, data augmentation defines a random vector θ. Arbitrary augmentation $\tau_{\theta }$ of image I is expressed as $I\to {\bar{I}}_{\tau_{\theta }}$. The introduction of data augmentation may cause the loss function not to converge. By combining the output results of the original data and the augmented data, it is regularized to improve the consistency of the data.

#### 3.4.1. View Mask

It is well known that the problems of view occlusion and image highlighting in multi-view reconstruction can irreparably affect the reconstruction results. Due to occlusion and wrong information in highlighted areas, the network will reconstruct a model that is far from the real scene. In order to resist the impact of view occlusion and highlighting on the reconstruction results in multi-view situations, we randomly generate a 2D mask $M_{\tau_{\theta_{1}}}$ to occlude part of the reference image. The 2D mask is then projected onto the neighboring frame image to block the corresponding area in the image. Assuming that the remaining region $I-M_{\tau_{\theta_{1}}}$ is not affected, the effective region between the original data and the augmented data output results are compared.

#### 3.4.2. Gamma Correction

Similar to JDACS [10], gamma correction utilizes a nonlinear method to adjust the brightness of the image. In this way, the impact of different views on output results due to changes in lighting conditions is simulated. $\theta_{2}$ is defined and $\tau_{\theta_{2}}$ represents the change of gamma correction to the original data.

#### 3.4.3. Color Transformation and Blurring

Nowadays, many image transformations can add some randomness to the color of the image. For example, it can make the color of the image fluctuate randomly, blur randomly, and add random noise to the image. This transformation enhances the unreliability of the data to a certain extent, therefore it needs to make the model robust to color transformation and improve the generalization ability of the model. $\tau_{\theta_{3}}$ represents the augmentation operation on the training data through color transformation. We change the properties of our image by changing its mean and variance. Specifically, we use the transforms function in Pytorch to achieve this operation. In addition, we add Gaussian noise to the image to simulate the influence of unfavorable factors on the image in real scenes. This strategy is used to improve the robustness of our algorithm. In conclusion, $\tau_{\theta }$ represents data augmentation. It is defined as:(9)$$\tau_{\theta }=\tau_{\theta_{1}}\circ \tau_{\theta_{2}}\circ \tau_{\theta_{3}}$$
where ∘ represents the combination of each augmentation function. Similar to JDACS [10], we implement this operation through the transforms function in Pytorch. We set our parameter factors as brightness = 1, contrast = 1, saturation = 0.5, hue = 0.5. For the size of the mask, we randomly select a point in the image and cut out 13 of the size of the original image.

### 3.5. Loss Function

In the depth estimation branch, corresponding to the reference image $I_{0}$, the real depth map is $D_{GT}$, the predicted depth map is $D_{E}$, and the depth estimation branch loss function is $L_{DE}$.
(10)$$L_{DE}=\sum_{k=1}^{3}\sum_{i=1}^{n_{k}}L_{i}^{k}\left(D_{E}-D_{GT}\right)$$

$L_{i}^{k}$ is the loss of the final depth estimation branch which is obtained by L1 loss calculation between the predicted depth map and real depth map in the $i_{th}$ iteration of the k-stage model.

In the data augmentation branch, it mainly includes 2D mask and image transformation operations. ${\bar{D}}_{\tau_{\theta }}$ represents the depth map predicted from the augmented reference image ${\bar{I}}_{\tau_{\theta }}$. The loss function of the data augmentation branch is $L_{DA}$.
(11)$$L_{DA}=\frac{1}{{\left\|M_{\tau_{\theta }}\right\|}_{1}}\sum_{k=1}^{3}\sum_{i=1}^{n_{k}}\left[\lambda_{DA_{1}}L_{i}^{k}\left({\bar{D}}_{\tau_{\theta }}-D_{GT}\right)+\lambda_{DA_{2}}L_{i}^{k}\left({\bar{D}}_{\tau_{\theta }}-D_{E}\right)\right]⊙M_{\tau_{\theta }}$$

$M_{\tau_{\theta }}$ represents the remaining part of the image after cropping. $\lambda_{DA_{1}}$ represents the weight between the depth map reconstructed by the data augmentation branch and the real depth. $\lambda_{DA_{2}}$ represents the weight between the depth map reconstructed by the data augmentation branch and the depth map reconstructed by the depth estimation branch. $\lambda_{DA_{1}}$ and $\lambda_{DA_{2}}$ are set to 0.8 and 0.2, respectively.

In summary, the total loss function of the model is Loss. It is defined as:(12)$$Loss=\lambda_{1}L_{DE}+\lambda_{2}L_{DA}$$

$\lambda_{1}$ and $\lambda_{2}$, respectively, represent the proportions of the depth estimation branch and the data augmentation branch in the total loss, and are initially set as $\lambda_{1}=0.8$ and $\lambda_{2}=0.1$. The proportion of $\lambda_{2}$ gradually increases as the training continues.

## 4. Experiments

In this section, DTU [1] and Tanks and Temples [2] datasets are utilized to evaluate our algorithm. The function of each part is shown through ablation experiments. Finally, a large number of experiments show that the algorithm is effective and accurate.

### 4.1. Datasets

DTU dataset [1]The DTU dataset is an indoor multi-view vision dataset in an experimental environment, which contains 128 different scenes and 49 views from different angles under different lighting conditions. The scanning of scene objects in this data set is carried out in a standard experimental environment, and they are all captured by cameras with the same trajectory. The DTU dataset is utilized to train and evaluate our network. The setup of most multi-view works is followed, providing 79 of the 128 scenes as the training set to provide real depth maps and 22 scenes as the test set to evaluate the 3D point cloud.Tanks and Temples dataset [2]

The Tanks and Temples dataset is a large-scale outdoor dataset for multi-view stereo vision. It utilizes a set of video frames in a real scene to represent the scan of the scene. This dataset contains 14 different scenes, divided into an intermediate dataset and advanced dataset. The network model obtained on the DTU dataset is utilized to evaluate our network on the Tanks and Temples dataset.

### 4.2. Implementation Details

The PyTorch deep learning framework is used to build our network model and train it on the DTU dataset. We use the PyTorch deep learning framework to build our network model and train it on the DTU dataset. For the input image, the image resolution is set to 640×512, the number of input images is set to *N* = 5 (one reference image and four source images). For the selection of source images, we adopt a robust training strategy similar to PVSNet. For each reference image, we randomly select 4 of the 10 best reference images. This process increases the diversity of training, thus improving the generalization ability. The number of iterations of the patchmatch algorithm at stages 1, 2 and 3 is set to 1, 2 and 2, respectively. During initialization, the depth layer is set as $D_{f}=48$. For local perturbations, the same parameter configuration as PatchmatchNet is set, such asN=7, $R_{2}=0.09$, $R_{1}=0.04$, $N_{3}=16$, $N_{2}=8$, $N_{1}=8$. In adaptive propagation, stages 1, 2 and 3 are set to 0, 8, 16 We set stages 1, 2 and 3 to 0, 8, 16 respectively (the last iteration of stage 1 needs to enter refinement, and therefore no propagation is required). For adaptive evaluation, we set $K_{e}=9$ at all stages. The parameters of the optimizer Adam are set to $\beta_{1}=0.9$,$\beta_{2}=0.999$, epochs = 9 and the learning rate = 0.001. During training, the batch size is 4 and training is on one Nvidia GTX 3090 GPU. In the network training calculation, when the image has a 640×512 resolution, the time required to process each image is 0.18 s. Compared with other algorithms, the calculation time is greatly shortened. In the test phase, the depth map of the test set is obtained through the trained model. The method of depth map construction is similar to MVSNet to obtain a point cloud.

### 4.3. Benchmark Performance

#### 4.3.1. Evaluation on DTU Dataset

In the evaluation phase on the DTU dataset, the original image size is 1600×1200, which is used as the input image. In the evaluation phase of the DTU dataset, we use the original image size is 1600×1200, as the input image size. The number of views is set to N=5. The range of sampling depth is set to $\left[425\;\mathrm{mm},935\;\mathrm{mm}\right]$.

The evaluation metrics are provided by the DTU dataset. As shown in Table 1, Gipuma performs best in accuracy and UniMVSNet performs best in comprehensive performance. The proposed algorithm in this paper is superior to other algorithms and is competitive in comprehensive performance. Our scheme reconstructs more point clouds in fine details, which also verifies the performance of our scheme on the integrity index, as shown in Figure 4. Due to adaptive propagation, our scheme can collect more accurate information from the boundary. Therefore, the effect of the proposed algorithm is better than that of CVP-MVSNet and CasMVSNet at the fine structure of the boundary. Both CVP-MVSNet and CasMVSNet use the idea of classification to divide the picture into different scales to extract its feature information, which can extract more feature information. However, 3D convolution is used in the construction of the cost volume, which greatly increases the calculation time and memory consumption. Our algorithm replaces 3D convolutions with adaptive cost aggregation, which reduces computation time and memory consumption.

#### 4.3.2. Calculation Time and Memory Consumption

In addition to the evaluation metrics on the dataset, calculation time and memory consumption are also important metrics used to evaluate the model. The proposed algorithm is compared with several state-of-the-art learning algorithms, such as CasMVSNet [23], UCS-Net [24], and CVP-MVSNet [22].

The comparison realizes the 3D model with the same resolution as the input image through 3D cost volume regularization. The comparison results are as shown in Figure 5. When the number of depth layers is fixed, the calculation time and memory consumption of all algorithms have a linear growth relationship with the image resolution (The image size 1920×1056 is taken as 100%. Due to the limitation of some algorithms on the image resolution, there is insufficient sampling for the high-resolution image. Therefore, it is impossible to reconstruct a better model). As shown in Table 1, the proposed algorithm outperforms other learning-based algorithms in calculation time and memory consumption. It is also competitive in performance.

#### 4.3.3. Evaluation on Tanks and Temples Dataset

The model is trained on the DTU dataset and evaluated directly on the Tanks and Temples dataset. For the input stage, the size of the input image is set to 1920×1056 and the number of views is set to N=7. Since the data in the Tanks and Temples dataset does not contain camera parameter information, the OpenMVG is utilized to calculate camera parameters and sparse point clouds.

As shown in Table 2, our algorithm outperforms PatchmatchNet [7] and is comparable to the top-scoring CasMVSNet [23] on the intermediate dataset. The proposed algorithm also has good performance on more complex advanced dataset. For the best performing UniMVSNet [9], the model is adjusted on the basis of DTU training, so that the model has better performance for complex outdoor scenes. In summary, our algorithm is more competitive and has good generalization performance compared to the algorithm based on 3D cost volume regularization due to its simple structure.

Since the Tanks and Temples dataset is aimed at outdoor scenes, it is more complicated to reconstruct the scene and lighting conditions. Our algorithm is only trained on the DTU dataset, and the reconstruction results on the Tanks and Temples dataset without any adjustments, are shown in Figure 6. Combining the results in Table 2 and Figure 6, the data enhancement method can improve the generalization ability of the model through methods such as image brightness adjustment and mask operation. It can reconstruct better results for complex reconstruction and complex lighting conditions.

### 4.4. Ablation Study

By performing ablation experiments, the effectiveness of each component structure of the proposed algorithm is proved. It is shown again that the following comparative experiments are all evaluated on the DTU dataset. This ablation study employs a controlled variable approach. When discussing the impact of one of the modules on network performance, other modules use the same parameter settings (except for the discussion module, other modules are included).

#### 4.4.1. Data Augmentation (DA)

A data augmentation module (DA) is added to our network structure. The data augmentation module can effectively reduce the model’s dependence on lighting conditions by changing the basic parameters of the image. The effect of highlighted regions on the reconstruction results is simulated by applying a mask to the image. In this section, the effectiveness of the data augmentation module is shown by comparison experiment, as shown in Table 3.

#### 4.4.2. Dynamic Interval d

In most learning-based algorithms, the depth sampling range $\left[d_{min},d_{max}\right]$ is determined and a depth layer is fixed within the depth range. Since the depth range of scene objects is continuous, this algorithm is simple but will lose a lot of useful information in the image.

The proposed algorithm utilizes the strategy of different intervals, which can divide the intervals according to the pixel interval in images of different scales. This can collect more useful information from coarse to fine. As shown in Table 4, our strategy can improve the performance of the model.

#### 4.4.3. Adaptive Propagation (AP) and Adaptive Evaluation (AE)

Adaptive propagation (AP) and adaptive evaluation (AE) are employed in our proposed model, which is contrasted with other fixed 2D offsets algorithms. The experimental results only with AP or AE are also evaluated, respectively. As shown in Table 5, the results validate that our adaptive propagation and adaptive evaluation modules can improve model performance.

#### 4.4.4. Number of Iterations of Patchmatch

In the training of the model, the network model does not need to perform adaptive propagation at stage 1. Therefore, the number of iterations is set to 1 at stage 1. In theory, the more iterations at each stage, the better the performance of the model. As shown in Table 6, experiments show that when the number of iterations increases to a certain value, the performance will not be improved. This indicates that the model has converged. Compared with algorithms that utilize a large number of neighborhood views for propagation, our proposed network model is embedded in a coarse-to-fine framework, which can speed up the convergence. In addition, the adaptive propagation in the network model and the initialization operation of the model can improve the model converge quickly.

#### 4.4.5. Number of Views

When evaluating on the DTU dataset, the number of standard views is set to N=5. Then the situation where the view number N=2,3,4,6 is evaluated. In multi-view reconstruction, more views can greatly improve performance. Table 7 validates the above-mentioned situation. With the increase in the number of views, the performance of the re-added model in terms of integrity and accuracy also improves; however, the memory consumption and calculation time also increase.

To sum up, the best performance scheme of our algorithm is that the network contains modules such as data augmentation, adaptive propagation, adaptive evaluation, and dynamic adoption interval. The number of reference frames is set to 5. The number of iterations in each stage is set to (1,2,2).

## 5. Conclusions

The matching algorithm based on data augmentation is proposed in this paper. The interval d is divided according to the pixel interval. The proposed algorithm also adopts adaptive propagation and adaptive evaluation modules. Compared with most learning-based algorithms, this algorithm consumes lower memory resources. The 3D cost volume regularization is replaced by adaptive evaluation; therefore, the model can have a faster processing speed.

In the traditional patchmatch method, the picture is directly divided into small patches, and its normal vector is calculated to connect each patch. However, using this method will greatly improve the calculation. Therefore, in the initialization stage of the depth map, we still adopt the method of plane scanning, which can save the consumption of computing resources. Due to the planar scanning method, part of the depth information will inevitably be lost in reconstruction. In planar scanning, the more depth layers are divided, the better the final reconstruction quality will be. Similarly, more computing resources and longer computing time are required to reconstruct each frame of image.

Although the structure and idea of this algorithm are simple, a large number of experiments on the DTU and Tanks and Temples datasets show that it can reconstruct a model with high integrity. Simultaneously it has low computing cost and good generalization ability. For the future, we hope to solve the defect of low accuracy without increasing the calculation consumption. This is conducive to applying the algorithm to small platforms with limited computing resources, such as wearable devices and mobile phones.

## Figures and Tables

**Figure 1 sensors-23-02729-f001:**
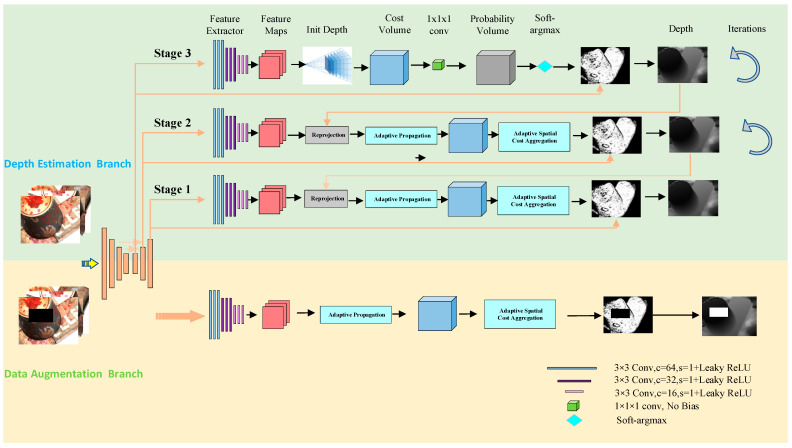
Network structure model. In the depth estimation branch, the depth map is initially initialized, and the depth map is obtained by calculating the cost body, 3D convolution, soft-argmax and other operations. Each stage can have multiple iterations. Upsampling the depth map can be used as the initial depth map for the next stage. The data augmentation branch and the depth estimation branch share weights in the feature extraction stage.

**Figure 2 sensors-23-02729-f002:**
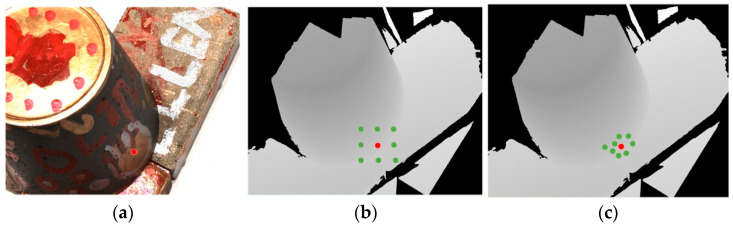
Example of adaptive propagation method: (**a**) means selecting a point on the original image; (**b**) conventional propagation method; (**c**) adaptive propagation method.

**Figure 3 sensors-23-02729-f003:**
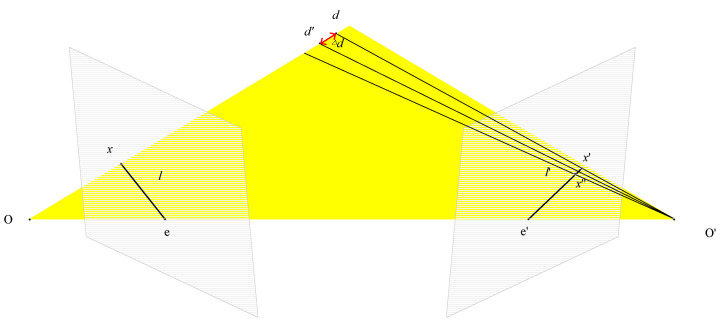
Calculation method of dynamic interval *d*.

**Figure 4 sensors-23-02729-f004:**
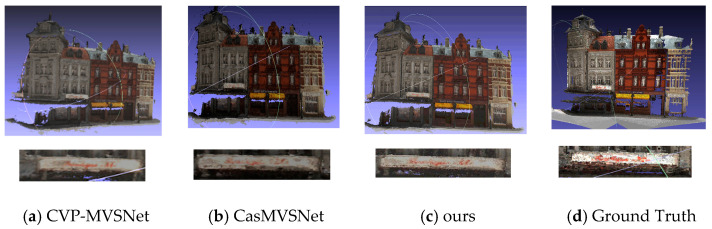
Scan 9 reconstruction quality comparison in DTU. (**a**) CVP-MVSNet reconstruction results. (**b**) CasMVSNet reconstruction results. (**c**) The reconstruction result of our algorithm. (**d**) Real scene in the dataset.

**Figure 5 sensors-23-02729-f005:**
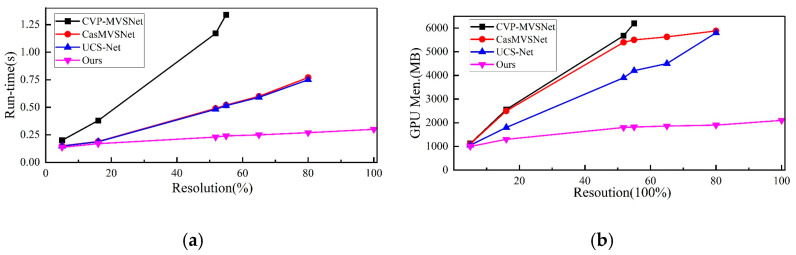
Comparing runtime and memory consumption. (**a**) The algorithm calculates the time required for each image. (**b**) represents the consumption of GPU memory resources.

**Figure 6 sensors-23-02729-f006:**
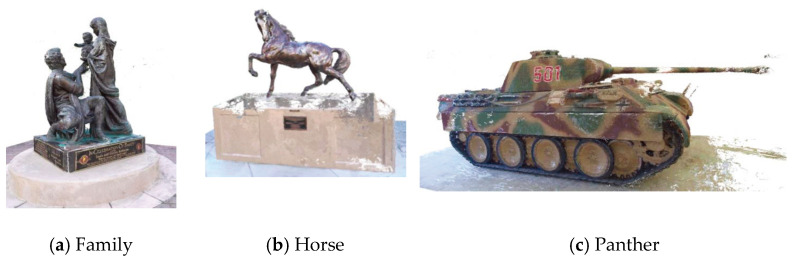
Demonstration of the point cloud reconstruction effect of the Tanks and Temples dataset. (**a**) Family point cloud reconstruction. (**b**) Horse point cloud reconstruction. (**c**) Panther point cloud reconstruction.

**Table 1 sensors-23-02729-t001:** Comparison of implementation results (the smaller the better).

Algorithm	ACC. (mm)	Comp. (mm)	Overall (mm)
ACMP [33]	0.835	0.554	0.695
Furu [34]	0.613	0.941	0.777
Gipuma [15]	**0.283**	0.873	0.578
COLMAP [16]	0.400	0.664	0.532
SurfaceNet [18]	0.450	1.040	0.745
MVSNet [3]	0.396	0.527	0.462
P-MVSNet [5]	0.406	0.434	0.420
R-MVSNet [4]	0.383	0.452	0.417
Point-MVSNet [35]	0.342	0.411	0.376
Fast-MVSNet [6]	0.336	0.403	0.370
AA-RMVSNet [28]	0.376	0.339	0.357
CasMVSNet [23]	0.325	0.385	0.355
CVP-MVSNet [22]	0.296	0.406	0.351
UCS-Net [24]	0.338	0.349	0.344
PatchmatchNet [7]	0.427	0.277	0.352
UniMVSNet [9]	0.352	0.278	**0.315**
Effi-MVS [36]	0.321	0.313	0.317
Ours	0.417	**0.272**	0.344

**Table 2 sensors-23-02729-t002:** Quantitative results on the Tanks and Temples dataset (F score, higher is better).

Algorithm	Intermediate Dataset									Advanced Dataset						
	Mean	Fam.	Fra.	Hor.	Lig.	M60	Pan.	Pla.	Tra.	Mean	Aud.	Bal.	Cou.	Mus.	Pal.	Tem.
COLMAP [16]	42.14	50.41	22.25	25.63	56.43	44.83	46.97	48.53	42.04	27.24	16.02	25.23	34.70	41.51	18.05	27.94
PointMVSNet [35]	48.27	61.79	41.15	34.20	50.79	51.97	50.85	52.38	43.06	-	-	-	-	-	-	-
UCS-Net [24]	54.83	76.09	53.16	43.03	54.00	55.60	51.49	57.38	47.89	-	-	-	-	-	-	-
P-MVSNet [5]	55.62	70.04	44.64	40.22	**65.20**	55.08	55.17	60.37	54.29	-	-	-	-	-	-	-
CasMVSNet [23]	56.84	76.37	58.45	46.26	55.81	56.11	54.06	58.18	49.51	31.12	19.81	38.46	29.10	43.87	27.36	28.11
ACMP [33]	58.41	70.30	54.06	**54.11**	61.65	54.16	57.60	58.12	57.25	37.44	**30.12**	34.68	**44.58**	50.64	27.20	37.43
VisMVSNet [27]	60.03	77.40	60.23	47.07	63.44	62.21	57.28	60.54	52.07	33.78	20.79	38.77	32.45	44.20	28.73	37.70
CVP-MVSNet [22]	54.03	76.50	47.74	36.34	55.12	57.28	54.28	57.43	47.54	-	-	-	-	-	-	-
Patchmatchnet [7]	53.15	66.99	52.64	43.24	54.87	52.87	49.54	54.21	50.81	32.31	23.69	37.73	30.04	41.80	28.31	32.29
AARMVSNet [28]	61.51	77.77	59.53	51.53	64.02	64.05	59.47	60.85	55.50	33.53	20.96	40.15	32.05	46.01	29.28	32.71
Fast-MVSNet [6]	47.39	65.18	39.59	34.98	47.81	49.16	46.20	53.27	42.91	-	-	-	-	-	-	-
EPP-MVSNet [37]	61.68	77.86	60.54	52.96	62.33	61.69	60.34	**62.44**	55.30	35.72	21.28	39.74	35.34	49.21	30.00	**38.75**
UniMVSNet [9]	**64.36**	**81.20**	**66.43**	53.11	63.46	**66.09**	**64.84**	62.23	**57.53**	**38.96**	28.33	**44.36**	39.74	**52.89**	**33.80**	**34.63**
Effi-MVS [36]	56.88	72.21	51.02	51.78	58.63	58.71	56.21	57.07	49.38	34.39	20.22	42.39	33.73	45.08	29.81	35.09
Ours	54.79	68.10	54.60	45.65	57.32	53.43	48.21	57.64	53.33	33.97	24.51	39.43	33.24	42.53	30.26	33.83

**Table 3 sensors-23-02729-t003:** Comparison of the impact of data augmentation on experimental results.

Algorithm	ACC. (mm)	Comp. (mm)	Overall (mm)
None	0.429	0.283	0.356
DA	**0.417**	**0.272**	**0.344**

**Table 4 sensors-23-02729-t004:** Effect of dynamic interval on experimental results.

Algorithm	ACC. (mm)	Comp. (mm)	Overall (mm)
Static d	0.429	0.283	0.356
Dynamic d	**0.417**	**0.272**	**0.344**

**Table 5 sensors-23-02729-t005:** Effect of adaptive propagation and adaptive evaluation on experimental results.

Algorithm	ACC. (mm)	Comp. (mm)	Overall (mm)
None	0.464	0.351	0.407
AP	0.437	0.293	0.365
AE	0.421	0.326	0.373
AP and AE	**0.417**	**0.272**	**0.344**

**Table 6 sensors-23-02729-t006:** Effect of different iterations on the experimental results.

Iterations	ACC. (mm)	Comp. (mm)	Overall (mm)
1,1,1	0.443	0.283	0.363
2,2,1	**0.417**	**0.272**	**0.344**
3,3,1	0.416	0.273	0.344
4,4,1	0.417	0.272	0.344
5,5,1	0.417	0.272	0.344

**Table 7 sensors-23-02729-t007:** Effect of different view numbers on the experimental results.

N	ACC. (mm)	Comp. (mm)	Overall (mm)
2	0.453	0.342	0.397
3	0.432	0.311	0.371
4	0.428	0.280	0.354
5	**0.417**	**0.272**	**0.344**
6	0.419	0.281	0.350

## Data Availability

The data presented in this study are available on request from the corresponding author. The data are not publicly available due to legal restrictions.

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
