# Peer review of "Multi-View Stereo Vision Patchmatch Algorithm Based on Data Augmentation"

_sensors, 2023, doi:10.3390/s23052729_

Round 1

Reviewer 1 Report

The performance of deep learning models depends on the amount and diversity of data used for training.  Data augmentation techniques are being used to transform the available data to generate new data.  

In the proposed work, the authors have used data augmentation technique to alleviate the problems associated with learning based Multi View Stereo such as extraction of wrong feature information.

comment-1

In the data input stage, data augmentation is used to enhance the image by changing the brightness, contrast.

Here the authors can explicitly write how the random removal of regions help? what is the purpose?

in depth estimation network, patchmatch is used for more feature extraction.

comment -2

Whether the proposed approach is freed from extraction of wrong feature information

Comment -3

the authors have described the advantages of proposed method, integrity, memory consumption, computing cost and good generalization.

here, authors can elaborate about limitations and how the accuracy can be improved can be highlighted

Reviewer 2 Report

Major:
Fig.1 - network structure is shown, but details for the reproduction of results are not available.

Minors:
l.62 and more
"dynamic interval d" - please change font or add ' ' for emphasis of 'd'

Fig.1
larger font size desired

l.211
1x1x1 convolution kernel with or without bias ?

l.242
please clarify naming and context of 'd' in this paper
"sampling interval 'd'", "dynamic interval 'd'", "interval 'd'"

l.273
Please add reference []

l.278
Please clarify used method for color transformation as well as random noise model

Eq.9
augmentation order is important, because nonlinear transformation occurs.
Please specify order of particular transformation as well as ranges of augmentation parameters in the table.

subsection 4.2
Processing time is important benchmark, also.

Reviewer 3 Report

The presented subject is interesting and is well written, but the following aspects must be presented with more details:

-discussion must be more elaborated: discussion must be presented based on the complexity of the object, illumination conditions, etc - it must be shown what are the results depending on these cases

-explain how were selected methods from the section 4.3 - results are not presented in section related work and is not clearly what are methods with best performances

-add also results obtained by the existing methods described in section related work and present their advantages and disadvantages 

Round 2

Reviewer 2 Report

ok

Reviewer 3 Report

Since all my comments are addressed, I recommend to publish the paper.

Just a small update: check alignment for paragraph at line 427.
